# Validity and Reliability of a Smartphone Application for Home Measurement of Four-Meter Gait Speed in Older Adults

**DOI:** 10.3390/bioengineering11030257

**Published:** 2024-03-06

**Authors:** Pei-An Lee, Clark DuMontier, Wanting Yu, Levi Ask, Junhong Zhou, Marcia A. Testa, Dae Kim, Gregory Abel, Tom Travison, Brad Manor, On-Yee Lo

**Affiliations:** 1Hebrew SeniorLife, Harvard Medical School, Boston, MA 02131, USAamylo@hsl.harvard.edu (O.-Y.L.); 2Beth Israel Deaconess Medical Center, Harvard Medical School, Boston, MA 02215, USA; 3VA Boston Healthcare System, Harvard Medical School, Boston, MA 02130, USA; 4Brigham and Women’s Hospital, Harvard Medical School, Boston, MA 02115, USA; 5Harvard T.H. Chan School of Public Health, Boston, MA 02115, USA; 6Dana-Farber Cancer Institute, Harvard Medical School, Boston, MA 02215, USA

**Keywords:** smartphone application, stopwatch, video, gait speed, four-meter walking test

## Abstract

The four-meter gait speed (4MGS) is a recommended physical performance test in older adults but is challenging to implement clinically. We developed a smartphone application (App) with a four-meter ribbon for remote 4MGS testing at home. This study aimed to assess the validity and reliability of this smartphone App-based assessment of the home 4MGS. We assessed the validity of the smartphone App by comparing it against a gold standard video assessment of the 4MGS conducted by study staff visiting community-dwelling older adults and against the stopwatch-based measurement. Moreover, we assessed the test–retest reliability in two supervised sessions and three additional sessions performed by the participants independently, without staff supervision. The 4MGS measured by the smartphone App was highly correlated with video-based 4MGS (r = 0.94), with minimal differences (mean = 0.07 m/s, ± 1.96 SD = 0.12) across a range of gait speeds. The test–retest reliability for the smartphone App 4MGS was high (ICC values: 0.75 to 0.93). The home 4MGS in older adults can be measured accurately and reliably using a smartphone in the pants pocket and a four-meter strip of ribbon. Leveraging existing technology carried by a significant portion of the older adult population could overcome barriers in busy clinical settings for this well-established objective mobility test.

## 1. Introduction

Short-distance gait speed tests—such as four-meter gait speed (4MGS)—are simple, powerful, widely accepted measures of objective physical performance for older adults [1,2,3]. The measurement of the 4MGS as a marker of frailty and mobility is recommended for general and subspecialty populations of older patients; for example, cardiology and oncology guidelines recommend the test for all patients undergoing cardiac surgery [4] and patients aged 65 years and older undergoing systemic cancer treatment [5]. However, barriers in busy clinical practices related to limited time, space, and staff have prevented more widespread implementation of gait speed tests. Additionally, the traditional stopwatch-based method for measuring gait speed requires another person’s assistance and is subject to human error and bias [6].

Standardized remote measurement of gait speed in home settings presents a potential alternative that can overcome barriers to measurement in clinical settings [7,8]. In addition, remote measurement creates the opportunity to collect data on mobility more frequently in home and community environments, where mobility matters most to maintain independence [8,9]. The home measurement of gait speed also obviates the need for travel to the clinic, which can be challenging for older adults who reside far away and/or have physical or financial limitations that make transportation difficult [10,11,12,13]. However, previous studies on remote gait speed measurement primarily utilized multiple advanced inertial measurement units (IMUs) attached to body parts such as the trunk and legs, requiring trained personnel for setup. There is, thus, a critical need to establish valid and reliable methods to complete the well-established protocols (e.g., 4MGS) for measuring gait speed in remote and unsupervised home-based settings.

Our team developed a smartphone-based application (App) that employs user-friendly, multi-media instructions to guide the user through a specific walking task with the phone placed in their pants pocket and, subsequently, identifies stride timing information from the data captured by the phone’s IMU [14]. To address the inconsistencies in various smartphone models, sensors, clothing, and environments, we tested multiple different phone models during our smartphone App’s development for enhanced compatibility. Our validation process also factored in statistical adjustments for different types of pants (tight, medium, or loose) and settings (laboratory or home), ensuring the App’s gait measurements remained accurate despite pocket tightness variations and environmental differences [14]. The smartphone App has also been tested on various older adults to ensure its usability across a wide range of technological proficiencies [14,15,16,17,18,19,20,21,22]. Moreover, previous studies suggest that older adults are interested in and willing to use smartphones to assist in the management of health [23,24,25]. This approach significantly differs from previous methods by eliminating the need for manual timing and the presence of a secondary observer. This innovation not only enhances the feasibility of conducting 4MGS tests at home but also aligns with the objective of enabling more frequent and contextually relevant mobility assessments outside clinical settings. Despite the prevalent use of smartphones for various measurements at home [26], there remains a challenge in accurately gauging gait speed using data from a single IMU sensor, particularly over short walking distances such as those used for 4MGS tests. To address this, we complemented the existing technology in smartphones with a 4-m ribbon and automatic detection of an individual’s turns at each end, thus providing the necessary spatial information to compute the 4MGS accurately. Our objectives were to (1) determine the validity of the smartphone App-based assessment of the 4MGS against gold standard video-based and stopwatch-based methods and (2) establish the reliability of the smartphone App-based assessment when used by older adults within their home, both with and without supervision.

## 2. Methods

### 2.1. Study Design and Participants

We conducted a design control verification and validation study by enrolling community-dwelling older adults (aged 65–90) residing in independent living centers. A total of twenty-one participants were enrolled in the study. To accommodate varying technological proficiencies, we offered two in-person trainings/observations on the use of the smartphone App by the study staff, ensuring all the participants were able to use the smartphone App. As part of our comprehensive assessment at baseline, we measured key demographic characteristics of the participants, including sex, age (in years), height (in meters), body weight (in kilograms), and ethnicity. Cognition was assessed using the Montreal Cognitive Assessment (MoCA). This widely employed test generates a global cognitive function score by assessing executive functions and memory. Physical function was assessed by the Short Physical Performance Battery (SPPB), which also provides a composite score derived from gait, balance, and chair-rising task performance. These tests were only conducted to help describe the functional ability of the recruited cohort in order to help interpret the generalizability of results (Table 1). We included those who had active Wi-Fi service in their homes and were able to use the smartphone App by themselves after training. Assistive devices were allowed if the participants normally used them when walking. We excluded individuals with two or more falls in the past six months; those with a history of ulcers, amputations, and/or other painful symptoms in their lower extremities; and individuals with neuromusculoskeletal diseases, neurological pathology, or moderate/severe dementia that prevented their ability to participate in the study. We kept our eligibility criteria broad in order to help ensure the recruitment of a cohort representing a relatively wide range of functional ability and technology proficiency. This study was approved by the Advarra Institutional Review Board (protocol ID: Pro00063604).

### 2.2. Smartphone Application (App)

Our team has developed an iOS smartphone App utilizing the phone’s inertial measurement unit (IMU) sensor for recording movement during walking while the phone is in the user’s pocket [14]. The smartphone App provides audio instructions throughout the test via the phone’s speaker. The App provides written instructions on the phone screen asking the participant to press a large “Start” button and then place the phone in their pocket. After approximately 10 s, the phone begins providing additional verbal instructions before giving a “Ready, Set, Go” command. Data capture begins with the word “Go”. This process helps to minimize potential delays caused by placing the phone into the pocket. After the audible “Go” command, a similar “Stop” command was provided after 45 s. Post assessment, the data were stored on the device and transmitted via Wi-Fi to a cloud database for offline storage and analysis.

The smartphone App provided a verbal signal “Go”, and the participants were instructed to walk along the line at their self-selected, comfortable pace, cross the end of the four-meter ribbon, make a turn, and walk back to cross the start of the four-meter ribbon. The participants walked back and forth multiple times over this four-meter walkway until the smartphone App signaled “Stop” at the end of the 45-s test. The participants were instructed to make a 180-degree turn immediately after crossing the endpoint of the four-meter ribbon, and the start of the turn was identified and used for the derivation of the four-meter gait speed.

### 2.3. Study Procedures and 4MGS Home Test

Each participant was asked to complete the 4MGS test in their home on five separate days within one week (Appendix A). The participants were advised to wear comfortable shoes and either pants or shorts with front pockets for each visit. In the first session, the study team traveled to the participant’s home, gathered validity data, and observed the participant’s ability to complete the 4MGS test from a standing start with the smartphone App. A pre-measured four-meter strip of ribbon was first placed on the ground in an area free of obstacles. The smartphone was placed in the participant’s pocket with the gait App open. Similar to standard clinical practice, the participants were allowed one or two practice sessions before data collection in order to warm up and to help mitigate any possible confounds due to unfamiliarity with the smartphone App or its instructions. Additionally, no instructions were provided regarding the direction of turns during the trial in order to ensure the conditions were as natural as possible. For this analysis, we obtained the time for the first instance of walking along the four-meter ribbon until the beginning of the first 180-degree turn. To concurrently measure the 4MGS via a gold standard, video cameras were set up to visually confirm the time (in seconds to two decimals) between when the smartphone App signaled “Go” and the participant’s foot crossed the end of the four-meter ribbon. To ensure consistency and minimize inter-rater variability, the same rater (L.A.) performed the stopwatch measurements for all the participants. Moreover, a different rater (P.-A.L.) derived the gait speed from video recordings, with no prior knowledge of the stopwatch measurements. The video was started well before the trial, and the beginning of the trial within the video was determined by identifying the frame corresponding to the audible “Go” command. The staff member also started the stopwatch at the point of the audible “Go” command.

In session two, the study team again traveled to the participant’s home to observe their independent self-administration of the 4MGS test using the smartphone App while walking along the four-meter ribbon, identical to the protocol above, including the video and stopwatch methods. For the following three unsupervised sessions, the participants used the smartphone App and four-meter ribbon to administer the test at home without study staff supervision (Appendix A). Each participant was asked to complete a total of five assessment trials within one week: two initial supervised trials to ensure the correct procedure and three subsequent unsupervised trials to evaluate the repeatability and participant adherence to the test protocol. This approach was instrumental in ensuring the accuracy of the smartphone App across diverse scenarios.

### 2.4. Derivation of 4MGS from Data Collected via Smartphone App and Gold Standard Methods

All the data and statistical analyses were performed using in-house programs within MATLAB (R2022b, MathWorks, Natick, MA, USA) and SPSS version 24 (SPSS Inc., Chicago, IL, USA). To derive the 4MGS from the smartphone App data, the tri-axial accelerometer data embedded in the smartphone were sampled at 100 Hz. Since the smartphone App could not measure the distance over which the patients walked, we determined the time between the start of the smartphone App signal (“Go”) and the beginning of the first turn at the end of the 4-m ribbon. We developed an automated method to identify turns to enable the four-meter time calculation using the local coordinate system embedded in the smartphone, which was described by direct cosine matrix (DCM). Turning 180° produces a large deviation in the orientation of the smartphone’s medial–lateral and anterior–posterior axes. The DCM data contained relatively small fluctuations during straight walking, with the values sharply changing in value and sign (with a zero crossing) during turning (Appendix A).

### 2.5. Statistical Analysis

We assessed the validity of the smartphone App for measuring the 4MGS by evaluating its agreement with the video-based measurement of the 4MGS. We first computed Pearson correlation coefficients (r), which measure the linear relationship between two datasets, and specified r greater than 0.90 as a very high correlation and 0.70–0.90 as a high correlation [27], aligning with previous studies that demonstrated high validity in various health and mobility assessments [28,29]. We then used the robust, non-parametric Passing–Bablok orthogonal regression, which is suitable for comparing different measurement methods while acknowledging measurement error [30]. This regression method does not assume a normal distribution of the data and is less affected by outliers, providing a slope and intercept that describe the relationship between methods. We calculated the magnitude of error among the gait speed data from the smartphone App and the video-based measurement, followed by generating a Bland–Altman plot to visualize this error as a function of the gait data [31]. The Bland–Altman plot is a graphical method that plots the difference between two measurements against their average, enabling the identification of any systematic bias and the limits of agreement. These analyses of agreement were repeated to compare the 4MGS measured by the smartphone App vs. the stopwatch method and the stopwatch method vs. the video-based method. We assessed the test–retest reliability of the smartphone App in measuring the 4MGS successively over a period of time using several intra-class correlation coefficients (ICCs). The ICCs evaluate the consistency of repeated measurements by comparing the variability of different measurements of the same subject to the total variation across all measurements and all subjects. We calculated the ICCs for the following three conditions: (1) for the smartphone App separately with supervision between two visits, without supervision among three visits, and between the average values of two visits with supervision and the average values of three visits without supervision, (2) for the video under supervision between two visits to address the intra-rater reliability of the video, and (3) for the stopwatch under supervision between two visits to address the intra-rater reliability of the stopwatch. In particular, we used a two-way mixed model (ICC 3, 1) for all conditions because it considers both the effects of individual subjects and the specific conditions under evaluation. We considered ICC values greater than 0.75 as excellent reliability and greater than 0.60 as good reliability [32].

## 3. Results

Twenty-one individuals were enrolled between September 2022 and February 2023. Three participants withdrew from the study due to technical difficulties, health conditions, or travel, respectively. The data from another three were excluded due to inadequate video capture of the gait speed. After these exclusions, we analyzed data from 15 community-dwelling older adults (mean ± SD age 77.67 ± 6.41 years, Table 1). We have included a detailed breakdown of the participant ages within our study population, assessing for any potential skewness in age, height, weight, body mass index (BMI), MoCA, and SPPB distribution. This analysis revealed that while there was a range in these demographics, the distribution was relatively homogeneous. None of the participants used any assistive devices. All 15 participants completed the supervised assessments in sessions one and two; however, two of the participants did not complete the unsupervised assessments in sessions three to five. The mean ± SD home 4MGS in the population (as per video-based measurement in the first session) was 0.87 ± 0.15 m/s for the supervised assessments and 0.80 ± 0.17 m/s for the unsupervised assessments; Table 1 presents the mean home 4MGS from the smartphone App, the video-based method, and the stopwatch-based method, measured with and without supervision. The data were reasonably well-approximated by a normal distribution.

### 3.1. Validity of 4MGS Measured by Smartphone App

In sessions one and two (under supervision), the 4MGS measured by the smartphone App and video-based methods were very highly correlated (r = 0.94, *p*-value < 0.001, Figure 1). Orthogonal regression analysis evaluating the association between the 4MGS measured by the smartphone App and the video-based methods revealed a strong agreement, with a slope of 0.95 and an intercept of −0.01 (slope 1 and intercept 0 indicates perfect agreement, Figure 1). The Bland–Altman plot showed minimal differences between the methods (mean difference 0.07 m/s) across a wide range of gait speeds, with a limit of agreement (±1.96 SD, the interval of the likely differences between two methods) of 0.12 m/s (Appendix A).

The 4MGS measured by the smartphone App and the stopwatch method were also highly correlated (r = 0.84, *p*-value < 0.001), as were the stopwatch and video-based method (r = 0.82, *p*-value < 0.001) during walking (Appendix A). Orthogonal regression analysis revealed that the slope between the smartphone App and stopwatch 4MGS was 0.86 m/s and the intercept was 0.04, and the slope between the stopwatch and video-based 4MGS was 1.02 and the intercept was −0.04 (Appendix A). The Bland–Altman plot showed that the limit of agreement between the smartphone App and stopwatch 4MGS was 0.12 m/s and between the stopwatch and video 4MGS was 0.24 m/s (Appendix A).

### 3.2. Reliability of 4MGS Measured by Smartphone App

The intra-class correlation coefficient (ICC, test–retest reliabilities) for the smartphone App 4MGS in sessions one and two (with supervision) was 0.85, in sessions three to five (without supervision) was 0.75, and between all supervised and unsupervised home sessions was 0.93 (Table 2). The ICCs for the video-based and stopwatch 4MGS in sessions one and two (with supervision) were 0.85 and 0.68, respectively (Table 2).

## 4. Discussion

We found excellent agreement in the measurement of the 4MGS between the smartphone App and video-based assessment in the homes of community-dwelling older adults, with minimal differences between the measurement methods across a range of gait speeds. Moreover, the test–retest reliability in the smartphone App measurement of the home 4MGS was high, even when the participants completed the test by themselves without supervision or assistance from the study staff. Together, these findings demonstrate the criterion validity and reliability of the smartphone App in remotely measuring the 4MGS in the homes of community-dwelling older patients in independent living facilities.

We present rigorous validations of the 4MGS as measured by a smartphone device in the homes of older adults. Previous studies primarily used smartphones or accelerometers attached to the trunk or pelvis [6,7,33,34,35,36,37]. Although this location closer to the body’s center of mass is believed to offer a more precise estimation of the gait speed compared to peripherally-worn sensors, our approach of placing the smartphone in the pants pocket demonstrated similar or better accuracy compared to prior work using a centrally-placed accelerometer [6,33]. Soangra and Lockhart found good agreement between their smartphone-based five-meter gait speed measurement and video-based capture in a laboratory setting, but their study was conducted on twelve healthy young adults (mean age 28 ± 4 years) as opposed to our study’s older demographic who completed the tests at home (mean age 78 ± 6 years) [33]. Moreover, compared to securing a wearable sensor to the trunk, using a smartphone may be more acceptable to older adults, [26] and placing it in the pocket obviates the need for additional equipment or training. Finally, other studies used digital health technology to remotely measure gait and mobility but lacked the standardized nature and interpretability of a gait speed test over a set distance [38]. In our research, the combination of a four-meter ribbon to guide the walking path of the participants with automatic turning detection at the end of the ribbon recapitulated the standardized 4MGS test with excellent validity and test–retest reliability in the homes of older individuals.

A comparison of the 4MGS measured by the smartphone App with the stopwatch—the most commonly used method in clinic and laboratory settings—further strengthened the validity of the smartphone App and revealed several of its advantages. The high correlation between the two methods provides convergent validity for the smartphone App, and the smartphone App even outperformed the stopwatch method when compared to gold standard video-based capture. The smartphone App showed a more accurate and precise measurement of the 4MGS compared to the stopwatch method, as indicated by the higher correlation between the smartphone App and the video-based assessment, the smaller differences between the smartphone App and the video-based method via Bland–Altman analysis, and higher test–retest reliability. A significant contributor to the increased variability of the stopwatch method can be traced to user error and inter-operator variability [6], stemming from person-to-person differences in reaction times, subjective judgment of when a participant reaches the endpoint of the course, and/or potential distractions that might affect the operator’s performance. The smartphone App’s automated detection of a participant’s turn at the end of the four meters not only removes the need for an additional individual (e.g., research staff or a caregiver [39]) to time the patient with a stopwatch but also mitigates the individual error and variation when the stopwatch timing is started and stopped.

The strong correlation between the smartphone App and video-based methods for measuring the 4MGS, supported by both the Pearson correlation coefficient (r = 0.94) and orthogonal regression analysis, underscores the practicality of using smartphones for remote gait speed assessments. This finding is particularly relevant for community-dwelling older adults who may have limited access to clinical settings. The convenience and accessibility of smartphone-based measurements can facilitate more frequent monitoring of gait speed. Furthermore, the ability to perform reliable gait assessments without the need for specialized equipment or personnel is expected to reduce barriers to routine mobility monitoring, supporting the proactive management of health conditions associated with gait speed. At least for older adults similar to those studied, the described smartphone approach may afford these advantages while still maintaining measurement validity.

The 4MGS derived from the smartphone App was, on average, slightly slower than the 4MGS measured by video-based capture. This may be due to the individuals beginning their 180-degree turn just after crossing the endpoint of the four-meter ribbon. This aspect of the methodology may lead to an underestimation of the gait speed, particularly in individuals who perform turns relatively slowly. Future efforts should thus aim to optimize the detection of the start of the turn to refine the accuracy of the gait speed measurement. Enhancements to the App’s algorithms to better detect a range of turning speeds and patterns would lead to even better validity and reliability for a broader range of users, including those with different gait patterns and turning speeds. To address this, further application and validation of the smartphone App are essential, particularly to ensure its accuracy to wider gait speed ranges.

Additional research is also warranted to improve the understanding, accessibility, and value of this type of home-based assessment. The current sample of this proof-of-concept study was small and relatively homogenous, which limits the generalizability of our results. Future work is thus needed to extend this work to other populations, including those with a greater representation of male individuals, older adults with more severe cognitive and/or physical limitations, and older adults with specific conditions such as cancer or cardiac disease. The feasibility of older adults adhering to longer-term serial home 4MGS testing should also be studied, given that our study evaluated daily measurements over just one week. Future research is also needed to (1) further optimize elder-friendly designs of the smartphone App, (2) explore minimal detectable changes in home-based gait speed measures, (3) address potential biases such as the Hawthorne effect (e.g., users trying to modify their gait because of their awareness of others observing them) [40], (4) identify the extent to which repeat testing induces learning effects, and (5) determine if anthropometric or other factors influence performance and/or measurement stability over longer periods of time. Finally, while the expressed goal of this study was to validate a home version of the 4MGS test, efforts are needed to identify functional assessments that provide similar information as the 4MGS yet are compatible with smaller homes that do not contain a 4-m walking path

## 5. Conclusions

The objectives of this study were to determine the validity of a smartphone App-based assessment of the 4MGS against gold standard methods and establish the reliability of the smartphone App-based assessment when used by older adults within their homes, both with and without supervision. The results demonstrated that the 4MGS can be accurately and reliably measured using the smartphone approach, as indicated by a strong correlation between the smartphone App and video-based measurements as well as high test–retest reliability over multiple days of testing. Leveraging the existing technology already present in devices carried by a significant portion of the older adult population could overcome barriers to measurement in busy clinical settings of a well-established objective test of mobility. Given its responsiveness to clinically meaningful changes [3,41], the home 4MGS measured by the smartphone App can be recorded at baseline and serially monitored to determine how older adult mobility changes through different disease states and in response to interventions. Accordingly, the validation of the home 4MGS via the smartphone App lays the groundwork for expanding the measurement of the 4MGS as a predictor and outcome in clinical practice and research involving older adults.

## Figures and Tables

**Figure 1 bioengineering-11-00257-f001:**
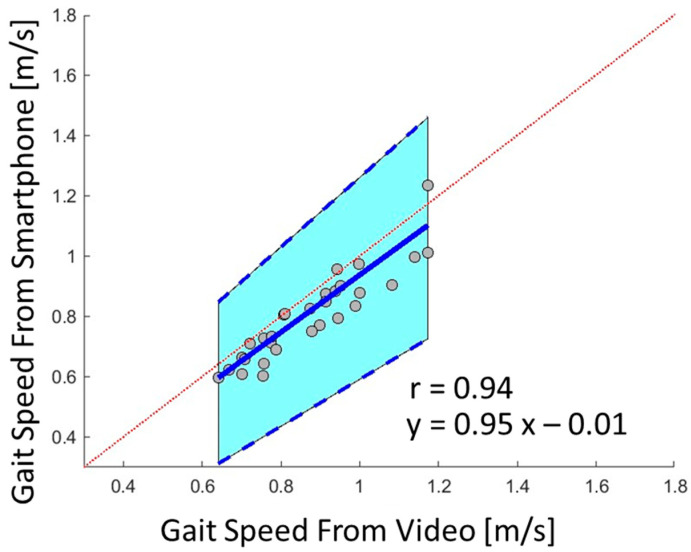
Passing–Bablok orthogonal regression evaluating the association and the relationship (r: correlation coefficient) between 4MGS measured by the smartphone App-based assessment and by the gold standard video-based method. The red dashed line represents the 45-degree line, indicating perfect agreement between two methods. The blue line represents the linear regression line that best fits the data points, and the blue dashed lines indicate the 95% confidence interval.

**Table 1 bioengineering-11-00257-t001:** Top: Demographic characteristic parameters of the participants presented by means ± standard deviations [maximums–minimums] if not specifically noted. Bottom: Four-meter gait speed (4MGS) recorded from the smartphone application (App), video, and stopwatch in older adults at home, with and without supervision.

	Demographic Characteristic Parameters					
Sex	15 women					
Age (years)	77.67 ± 6.41 [88.00–67.00]					
Height (m)	1.62 ± 0.06 [1.69–1.47]					
Body weight (kg)	70.00 ± 15.26 [92.00–43.50]					
Body mass index (BMI)	26.78 ± 5.96 [37.36–17.21]					
Ethnicity	13 white or Caucasian, 1 black or African American, and 1 other group					
MoCA	25.93 ± 2.58 [29.00–20.00]					
SPPB	9.67 ± 2.38 [12.00–5.00]					
	**Four-Meter Gait Speed (4MGS)**					
**With Supervision**						
	Smartphone App		Video		Stopwatch	
	Day 1	Day 2	Day 1	Day 2	Day 1	Day 2
Gait Speed (m/s)	0.81 ± 0.17 [1.23–0.60]	0.79 ± 0.13 [1.01–0.60]	0.88 ± 0.15 [1.17–0.70]	0.87 ± 0.15 [1.17–0.64]	0.94 ± 0.26 [1.55–0.68]	0.84 ± 0.14 [1.17–0.61]
**Without Supervision**						
	Smartphone App					
	Day 3		Day 4		Day 5	
Gait Speed (m/s)	0.82 ± 0.15 [1.20–0.61]		0.79 ± 0.21 [1.14–0.30]		0.80 ± 0.16 [1.16–0.61]	

MoCA: Montreal Cognitive Assessment; SPPB: Short Physical Performance Battery.

**Table 2 bioengineering-11-00257-t002:** Test–retest reliability of home 4MGS measured by smartphone application (App), video-based, and stopwatch methods.

	ICC	*p*-Value	95% CI
**Home assessment with supervision between two visits**			
Smartphone App	0.85	<0.001	0.62–0.95
Video	0.85	<0.001	0.62–0.95
Stopwatch	0.68	<0.001	0.27–0.88
**Home assessment without supervision among three visits**			
Smartphone App	0.75	<0.001	0.49–0.91
**Home assessment with and without supervision**			
Smartphone App	0.93	<0.001	0.77–0.98

ICC: Intra-class Correlation Coefficient.

## Data Availability

The data presented in this study are available on request from the corresponding author.

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
