# Peer review of "Validity and Reliability of a Smartphone Application for Home Measurement of Four-Meter Gait Speed in Older Adults"

_bioengineering, 2024, doi:10.3390/bioengineering11030257_

Round 1
Reviewer 1 Report
Comments and Suggestions for Authors
The purpose of this study was to evaluate the validity and reliability of this smartphone App-based home 4MGS assessment. The topic is quite interesting. Despite the brief length of this paper, its content is quite comprehensive. Before possible publication, some minor issues must be addressed by the authors.
1. To make the paper more interesting and provide broad knowledge to the reader, the author must add a mathematical background related to the static analysis used.
2. Apart from that, it is recommended to briefly explain unfamiliar terms that are used as the main parameters in the analysis, such as what is meant by intra-class correlation coefficients (ICC).
3. The author needs to add a scheme or at least a flowchart that explains the design research used
4. What is the reason for the narrow speed range of 0.6 to 1.2 m/s? The gait speed of older individuals in certain countries can reach up to 1.5 meters per second. Is the application suitable for individuals with speeds that fall outside the range examined?
5. What is the appropriate R-value to measure the validity of a smartphone application? Authors are required to add references to support their claims. In several studies with different subjects, the value r = 0.94 is still "not valid" considering computer binarity
6. Comparing the measurements of 4MGS obtained through the smartphone App and stopwatch method, the value of r decreases. For what reason? Please provide a more comprehensive explanation.
Author Response
We appreciate the reviewer’s time and positive comments on this manuscript. Below please find our specific responses to each comment. The revised content is highlighted in light yellow.

Reviewer 2 Report
Comments and Suggestions for Authors
Introduction
The introduction highlights the importance of the four-meter gait speed (4MGS) test in assessing frailty and mobility in older adults. However, the authors should briefly mention how the smartphone application differs from previous methods and devices in order to address the previous limitations and highlight the gaps that the present study wish to fulfill.
The authors state the need for remote measurement of gait speed but do not provide evidence or references supporting this need, especially in the context of its effectiveness compared to traditional methods. Please, provide more info.
Furthermore, the authors assume that older adults are comfortable with using smartphones and applications, however, this might not be collectively true and could therefore affect the applicability of the provided solution. Please give more rationales.
In the introduction, a brief discussion of limitations of using a smartphone App for measuring gait speed might be beneficial. For example, variability in smartphone models and their sensors might affect the consistency of measurements might be elucidated.
It might be highly helpful for readers to acknowledge the limitations and challenges in using a 4-meter ribbon in a home setting, considering space constraints in some homes. Please provide, previous approaches that you found in the literature and previous researches, and how previous authors tried fulfil these gaps.
Methods
The authors need to provide more information about the test procedure. This includes details about warm-up routine, and instructions given to participants regarding the direction of turns (whether it was random or specified).
Furthermore, clarity is needed on whether participants were instructed to wear shoes or perform the test barefoot. This could significantly impact gait speed and should be standardized across participants.
There is a lack of detail on how the smartphone App was initiated. It is unclear whether the "Go" command was activated at the beginning of the walk with the phone already in the pocket, or if there was a delay between pressing "Go", placing the phone in the pocket, and starting the walk. The potential impact of this delay on the results should be addressed.
In the unsupervised tests, the procedure for stopping the app is not mentioned. It would be beneficial to know how and when participants were instructed to stop the app to ensure consistency in data collection.
The paper should clarify whether the 180-degree turn was made exactly at the end of the 4 meters, and whether the data cutoff for analysis was before or after this turn. These are important information for data analysis and results usage.
Information is also needed on the synchronization points for the video analysis, stopwatch, and smartphone App's "Go" command. Understanding the exact moment of synchronization (e.g., first step, rapid movement before start, when the first step crossed a line) is crucial for data accuracy and results interpretation.
The number of familiarization trials provided to participants should be mentioned. This is important to ensure that participants were comfortable with the test procedure.
Controlling for the time of day when the tests were conducted is also essential, as it can influence the physical performance of older adults.
The authors should provide more information about the cognitive and physical function tests used: the Montreal Cognitive Assessment (MoCA) and the Short Physical Performance Battery (SPPB), and how these variables relates with the specific test.
Detail on the number of test trials performed by each subject is required for a complete understanding of the study design.
The authors also mention that assistive devices were allowed if normally used, but it doesn't specify how the use of these devices was accounted for in the data analysis, which could influence gait speed measurements and the start for the unsupervised session. Please be more specific and detailed.
There is an assumption that all participants are equally skilled at using the smartphone App. Variability in technological proficiency among older adults may impact the consistency and accuracy of the data collected. How was this controlled?
The authors should elaborate more on the participant recruitment process, total number of participants, and how the study accounted for varying technological proficiency among participants.
Statistical analysis
It is important to clarify whether the stopwatch used for measuring 4MGS was always operated by the same rater. Consistency in who performs the measurement is crucial to minimize inter-rater variability. How was this statistically addressed?
Similarly, information is needed on whether the video analysis was always conducted by the same rater. This consistency is essential for reliable data analysis. How was this addressed? Any reliability data in intra-rater or inter-rater reliability?
The paper should explain why specific reliability procedures were chosen for calculating the Intra-class Correlation Coefficients (ICCs) of the APP. Additionally, it should specify the type of ICC used (e.g., mixed, fixed) as this influences the interpretation of the results.
Inclusion of the Standard Error of Measurement (SEM) and the Minimal Detectable Change at 95% confidence (MDC95) would provide a more comprehensive understanding of the measurement's precision and the smallest change that can be considered above the measurement error.
Information on whether the data were normally distributed is crucial. This affects the choice of statistical methods and the interpretation of results.
Subsequently, the suggestion to perform an Analysis of Variance (ANOVA) should be taken into consideration. ANOVA could detect any learning effect or ceiling effect, particularly important in repeated measures over the test session.
It would be beneficial if the authors could provide justification for the selection of the selected statistical methods. For example, why was Passing-Bablok regression chosen over other regression methods? Please provide also for the other analysis.
Results and Discussion
The sample appears to be potentially unbalanced in terms of age. The paper should provide a more detailed analysis of the demographics to understand if there's a skew in the participant recruited. An unbalanced sample in terms age could impact the generalizability of the reliability and validity, especially with a small sample size.
It would be beneficial for the paper to include the BMI of participants, as BMI can influence gait speed and may be an important confounding factor to consider in the analysis. How was this controlled in the analysis?
More comprehensive information about the MoCA and SPPB assessments used should be provided. This includes the rationale for their use, how they were administered, and how their scores were interpreted in the context of this study.
I would suggest the authors to include all figures (such as Bland-Altman plots and Passing-Bablok regression graphs) directly in the manuscript. Visual representation of data can significantly enhance understanding and provide a clearer picture of the results.
The difference in gait speed between supervised and unsupervised assessments is important. The authors should statistically investigate and provide more information into the possible reasons for this difference.
As the correlation between the smartphone App and video-based methods are strong, as indicated by the Pearson correlation coefficient and orthogonal regression analysis, it would be helpful to have more discussion on the practical implications of these findings.
I would suggest the authors to also provide an overall specific section that highlight the practicality of the results.
Comments on the Quality of English LanguageMinor editing of English language required
Author Response
We appreciate the reviewer’s time and comments on this manuscript. Below please find our specific responses to each comment. The revised content is highlighted in light yellow.

Round 2
Reviewer 2 Report
Comments and Suggestions for Authors
I thank the author's for addressing all my comments. I have no more comments.
Comments on the Quality of English LanguageMinor editing
Author Response
Reviewer 2:
We appreciate the reviewer’s time and comments on this manuscript.
Comments and Suggestions for Authors:
- I thank the author's for addressing all my comments. I have no more comments.
Response: Thank you for the positive comment.
Comments on the Quality of English Language:
Minor editing
Response: The quality of the English Language has now been improved.